# Do Framed Mental Health Messages on Social Media Influence University Students’ Motivation for Physical Activity?

**DOI:** 10.3390/ijerph18168671

**Published:** 2021-08-17

**Authors:** Georgia Gilbert, Chloë Williamson, Justin Richards, Taya Annabelle Collyer, Paul Kelly

**Affiliations:** 1Edinburgh Medical School, The University of Edinburgh, Edinburgh EH16 4TJ, UK; 2Physical Activity for Health Research Centre (PAHRC), The University of Edinburgh, Edinburgh EH8 8AQ, UK; chloe.williamson@ed.ac.uk (C.W.); p.kelly@ed.ac.uk (P.K.); 3Faculty of Health, Victoria University Wellington, Wellington 6140, New Zealand; justin.richards@vuw.ac.nz; 4Peninsula Clinical School, Monash University, Frankston, VIC 3199, Australia; taya.collyer@monash.edu

**Keywords:** physical activity, message framing, social media, mental health, health promotion

## Abstract

Message framing has been used as a strategy for promoting physical activity (PA) in university students, but the effectiveness of gain-framed (GF), or loss-framed (LF) messages is variable. This study aims to investigate the effects on motivation and PA behaviour of framed messaging on social media in university students. Gain- and loss-framed messages communicated the mental health outcomes of PA. A three-arm feasibility study (*n* = 148) collected pre-post intervention online questionnaire responses to assess motivation for PA, exercise, active travel, and PA levels, in response to the messaging intervention on Facebook. Both GF and LF messages effectively increased average motivation for PA in comparison to controls (GF by 0.3 (on a 7-point Likert scale), 9% [95% CI: 3–17%], *p* = 0.007, LF by 0.3, 10% [CI: 3–18%], *p* = 0.005). Average motivation for exercise increased in comparison to controls (GF by 0.6, 16% [95% CI: 6–26%], *p* = 0.001, LF by 0.5, 14.6% [95% CI: 5–26%], *p* < 0.001). Average motivation for active travel increased in comparison to controls (GF by 0.7, 18% [95% CI: 8–29%], *p* < 0.001, LF by 0.6, 19% [95% CI: 8–30%], *p* < 0.001). No meaningful differences between GF or LF messages were observed. Framed messages regarding mental health outcomes of PA delivered via social media could be effective for increasing PA motivation in university students. However, based on our results there is no gain- or loss-framed advantage.

## 1. Introduction

Physical activity (PA) is associated with the prevention and management of many chronic diseases including cardiovascular, metabolic, and mental disorders [1]. Existing PA levels have been shown to avert 3.9 million (95% CI 2.5–5.6) premature deaths annually [2].

University students are a specific population of interest within PA research for two main reasons. Firstly, approximately 70% of UK university students are not active enough to meet government recommendations [3]. This may be due to academic pressure and socio-cultural norms in the university lifestyle [4]. Such socio-cultural norms include up to 8-h per day studying, sitting talking with friends, playing computer games and watching television [5]. Secondly, PA behaviours in this population may track into lifelong active habits and future health benefits [6]. Therefore, developing interventions that promote PA behaviour in students is a research priority.

Communication approaches such as messaging form a category of interventions that may aid PA promotion [7,8]. One such approach is framed messaging, which has been used as an effective strategy for impacting various outcomes such as awareness, knowledge, motivation, and in turn, PA behaviour [8]. Self-determination theory (SDT) has been recommended as a moderator to increase and sustain motivation to be active [9]. SDT is a theory of motivation which suggests that regulation of a behaviour is on a scale from non-self-determined (amotivation, introjected, and external regulation) to completely self-determined regulation (intrinsic regulation) [10], highlighting the importance of motivation as a precursor to behaviour.

Messages can be framed in terms of “gains” or “losses”. Gain-framed (GF) messages highlight the benefits of being active, while loss-framed (LF) messages highlight the consequences of being inactive. Framed messaging is founded upon Prospect Theory, which states that the risk associated with a given behaviour is influential on the decision to partake in said behaviour [11]. Rothman and Salovey [12] theorised that GF messages are more effective for behaviours perceived as low risk (e.g., PA) and LF messages are more effective for behaviours perceived as high risk (e.g., smoking). A number of systematic reviews support this theory in the context of promoting PA [13,14], however, conclusions are limited by the low methodological quality and heterogeneity of the original research articles included in the reviews. There is therefore a need for rigorous studies to investigate whether GF or LF messaging is more effective in this context. It has also been concluded that the literature on framed messaging for PA promotion contained contradictory findings and further research is required to find effective messaging strategies [9].

Message framing studies lack consistency in their use of control groups [14]. Some studies use no message control groups [15,16] which cause difficulty in distinguishing if the content of the message, or simply regular contact, caused the behaviour change. This presents a gap in the literature for studies that include a message control group. Important limitations are also present in previous studies. For example, a previous study showed beneficial effects of both GF and LF messages in a student population, via email delivery [16]. However, this study had initial demographic group differences and a short intervention period of only three weeks, presenting further gaps in the literature.

Existing evidence supports the use of GF messages, particularly those focusing on mental health benefits [8], despite physical health benefits being at the forefront of most PA messages [17]. Mental health promotion is an important area of research, particularly in the student population. Between 2006 and 2016 there has been an approximately fivefold increase in mental illness in UK first-year university students [18]. Despite this, the use of messages focussing on mental health outcomes is an under-researched area. PA promotion efforts are warranted in this university population, particularly when targeting outcomes such as motivation [8].

It is estimated that 63% of UK 16–24-year-olds consider Facebook to be their main social media platform [19]. Facebook has been promoted for use in health interventions due to previous success in attracting and retaining participants [20]. Facebook provides us with an exciting opportunity to improve health through technology, due to its vast reach and cost-effectiveness [21]. Therefore, Facebook is a promising potential delivery platform for messages to university students.

While there have been previous studies assessing GF and LF messages, there is an evidence gap in studies that include a control group, use social media, and analyse a student population. This study aims to investigate the effects on motivation and PA behaviour of university students in response to social media messaging framed around mental health outcomes.

## 2. Materials and Methods

### 2.1. Participant Recruitment

Participants were recruited from UK universities by convenience sampling through course announcements, emails, flyers, posters, and social media platforms. A prize draw of GBP 50 was advertised to maximise recruitment. Inclusion criteria required participants to be aged 18–26, a student at a UK university, and have a Facebook account. This age range was chosen to represent the majority of the student body. Ethical approval was attained (on 30 November 2018) from the institution’s Research Ethics Committee. All participants provided informed consent electronically.

### 2.2. Instruments

Sample messages and questionnaire options were trialled before the intervention began. Feedback indicated that messages should be sent twice daily to maximise the likelihood of being seen by participants. Care was taken in message development to frame messages effectively, in line with examples provided by Li et al. [22]. Information contained within each message was evidence-based from peer-reviewed journals, sourced from the Physical Activity Guidelines Advisory Committee [23] report. Participants in the control group received messages involving random health facts, unrelated to PA (examples shown in Table 1).

### 2.3. Procedure

This study followed a three-group randomised control design with two intervention conditions and one control condition. Eligible participants were randomised into GF, LF, or control groups, using an online random list generator [24]. Outcome data were collected via an online questionnaire (Appendix A) at two time points (baseline and end of intervention).

A link was circulated around the university for recruitment, consent, and collecting demographic and baseline data. This took place over a two-week period. Group allocation then occurred, and the five-week messaging intervention began (4 February–10 March 2019). This time period was chosen to assess a longer delivery protocol than previously tested [16], without undue burden on participants. The length of time between baseline data collection and intervention start date differed, but the inherent bias was avoided due to random group allocation. Participants received two messages per day (9 a.m. and 1 p.m.). Evidence supports sending short messages to young adults at times when there is an opportunity to act on them, such as near morning or afternoon work breaks [8]. Participants received a notification that the message was in their Facebook group. Researchers could check how many participants had viewed the message on each group, which was used to assess engagement. Participants in all three groups expected to receive messages, however, information about message framing and group allocations were concealed from participants. Figure 1 depicts the flow of participants through each of the stages.

### 2.4. Outcomes

Motivation for PA had three components: ‘Motivation for PA’; ‘Motivation for exercise’, and ‘Motivation for active travel’, from the four-domain model for PA [25]. Motivation outcome measures were measured using responses to statements such as, “I feel motivated to participate in exercise” on a 7-point Likert scale. A score of 1 indicated, “strongly disagree”, and 7 indicated, “strongly agree”. An adapted version of the Behavioural Regulation in Exercise Questionnaire 3 (BREQ-3) [26] was used. It was adapted to prioritise candidate domains of interest, reduce participant burden, and evaluate all aspects of PA, not just exercise. The PA level was assessed using the IPAQ-S to assess frequency and duration of weekly walking, moderate PA, vigorous PA, and total PA. Using published IPAQ data processing guidelines, the total physical activity in MET-minutes/week was calculated [27].

### 2.5. Data Processing and Statistical Analysis

On completion of the study period, survey responses were exported, cleaned, and stored in Microsoft Excel. Data were analysed in Stata version 15. The IPAQ data were processed according to steps provided in the IPAQ scoring protocol [27]. Message engagement for each group was calculated by combining the number of views on each message post and calculating a mean for each week of the intervention.

Descriptive statistics were calculated for demographics. The means and standard deviations were calculated for all motivation outcomes and IPAQ scores. Motivation for PA, exercise, active travel and PA level (MET minute/week) were analysed by analysis of covariance (ANCOVA), adjusted for baseline values as is recommended for trials with baseline and follow-up measures, to avoid confounding via regression to the mean [28]. Motivation for PA and active travel violated the heteroskedasticity assumption. These were log-transformed after scrutinising residual plots. Beta coefficients (β) represent effect sizes and are estimates of the difference between control and intervention conditions. α = 0.05 for all hypothesis tests.

## 3. Results

### 3.1. Demographic

Demographic data were analysed for N = 147 participants to assess between group differences, as seen in Table 2.

All groups had a larger proportion of female participants than males (75.5% female overall). The majority were Caucasian (88.4%), in line with the overall demographics of UK universities [29], and most participants attended Scottish universities (85.5%). There were no significant differences between the control, GF and LF groups in age (*p* = 0.66), sex (*p* = 0.23), ethnicity (*p* = 0.43) and university (*p* = 0.36).

Pre-post intervention differences in motivation for PA, exercise, active travel and PA level were calculated, as seen in Table 3. There was some variation in baseline values between groups for all outcomes. This was accounted for in the analysis by adjusting for these baseline values.

### 3.2. Motivation for PA

Adjusting for baseline score, those who received GF messages had average final PA motivation scores 9% higher (95% CI: 3 to 17%) than controls (*p* = 0.007). Adjusting for baseline score, those who received LF messages had average final PA motivation scores 10% higher (95% CI: 3 to 18%) than controls (*p* = 0.005). There was no statistically significant difference in study-end in motivation for PA between the GF and LF groups (*p* = 0.87).

### 3.3. Motivation for Exercise

Adjusting for baseline score, those who received GF messages had average final motivation for exercise scores 16% higher (95% CI: 6 to 26%) than controls (*p* = 0.001). Adjusting for baseline score, those who received LF messages had average final motivation for exercise scores 15% higher (95% CI: 5 to 26%) than controls (*p* < 0.001). There was no statistically significant difference in study-end motivation for exercise between the GF and LF groups (*p* = 0.93).

### 3.4. Motivation for Active Travel

Adjusting for baseline score, those who received GF messages had average final motivation for active travel scores 18% higher (95% CI: 8 to 29%) than controls (*p* < 0.001). Adjusting for baseline score, those who received LF messages had average final motivation for active travel scores 19% higher (95% CI: 8 to 30%) than controls (*p* < 0.001). There was no statistically significant difference in study-end motivation for active travel between the GF and LF groups (*p* = 0.90).

### 3.5. PA Level (Total MET Minutes per Week)

Adjusting for baseline IPAQ scores, there was no statistically significant difference in the post IPAQ score for the GF group (β = 0.02, 95% CI: −0.06 to 0.9), compared to controls (*p* = 0.6). Adjusting for baseline IPAQ scores, there was no statistically significant difference in the post IPAQ score for the LF group (β = 0.01, 95% CI: −0.06 to 0.09), compared to controls (*p* = 0.75). There was no statistically significant difference in IPAQ scores between GF and LF groups (*p* = 0.9).

### 3.6. Message Engagement

The GF group had the highest total mean message views throughout the intervention (M = 36.2) and the LF group had the lowest (M = 32.4). All groups mean message views peaked in week 2 (GF M = 40.9, LF M = 35.6, control M = 35.2) and declined in week 3 (GF M = 34.8, LF M = 30.9, control M = 32.2), remaining steady throughout the rest of the intervention, as seen in Figure 2.

## 4. Discussion

### 4.1. Principle Findings

This study aimed to assess if framed messages regarding mental health outcomes of PA were effective in increasing motivation for PA and the PA level of university students via Facebook delivery. We found that students who received both GF and LF messages increased their motivation for PA, exercise, and active travel, more than those who received random non-PA health facts. There were no significant differences between GF or LF messages on any outcome measure. The PA level did not change significantly in any of the groups. Message engagement was highest in the GF group but followed similar patterns for all three groups over the course of the intervention.

### 4.2. Comparisons to Literature

With regards to motivation for PA results, this study shows that mental health-based messages about PA have a positive effect on PA motivation in a student population. This finding is in keeping with other literature which has found positive effects of mental-health based messaging in young adults, specifically in relation to motivation and self-efficacy outcomes [8].

With regards to PA level results, our finding that there was no significant effect on PA level contrasts to previous literature which found that GF messages increased PA level significantly and more than LF messages [13,14].

This study also found there were no significant framing effects. This is in line with existing literature demonstrating contradictory findings with regards to GF or LF advantage [9,30]. In agreement with this finding, Parrott, Tennant, Olejnik, and Poudevigne [16] found that GF and LF messages are both effective in promoting exercise intention in a student population. Our study builds on and strengthens this evidence by demonstrating similar findings in a randomised controlled trial design where the previous study did not include a control group [16]. In a study that found GF messages to be more beneficial in a cardiac population, weaknesses such as small sample size (*n* = 49) were present [15], reducing the statistical power of the results. Although our results did not find any significant framing effects, the larger sample size has more statistical strength. Conversely, in a study that found negatively framed messages to be more effective, solely post-test assessments were carried out [31], which is less reliable than the pre-post intervention assessment in this study.

### 4.3. Plausible Explanations

Our results suggest that increased motivation in the LF and GF groups was due to increased knowledge about the mental health benefits of PA. This could be due to The Elaboration Likelihood Model which suggests that persuasive messages convey information that may cause attitude changes in the audience [32].

Many of the messages in this study were focused on short term mental health benefits (e.g., immediate improvements in mood or stress levels). The benefits of these messages may relate to social marketing techniques that allow the recipient to ‘buy into’ appealing and immediate outcomes [8]. Affective-based messages (e.g., immediate mood changes) have been shown to increase PA level more, in comparison to ‘instrumental’ messages (e.g., PA can help maintain a healthy weight) [33].

Despite changes in motivation, no effect on PA behaviour was detected in the GF or LF groups compared to the control group. It is also not clear why our results differ from those of Gallagher and Updegraff [13] and Latimer, Brawley and Bassett [14]. This may be explained by the cohort being too active at baseline or the 5-week intervention being too short to see meaningful effects on PA level. However, it can be argued that behaviour change is not the immediate purpose of messaging. Perhaps the focus should be on influencing proximal variables such as awareness, knowledge, motivation and intention, in order to promote long-term behaviour change [34].

### 4.4. Strengths

Formative research prior to developing carefully framed messages was undertaken, as recommended by Williamson, Baker, Mutrie, Niven and Kelly [8]. This was carried out in order to avoid conflating messaging styles of GF and LF versus positive and negative framing. Group randomisation avoided selection bias and minimised the effects of potential confounding variables. No missing data was attained, and attrition rates were low (3.9%). Our study focused on an under-researched and critical population group for PA promotion [6]. It was also a successful use of a low-cost intervention that used a social media platform that is ubiquitous and readily available for future intervention.

### 4.5. Limitations

This study has important limitations. Firstly, the recruitment method resulted in a highly active sample. 46.3% of participants were deemed to have high PA status, categorised by IPAQ responses. Therefore, findings were impacted by ceiling effects. It is possible that these active individuals did not have much scope to become more active, and future research could investigate the effects of GF and LF messages on low or inactive groups. Additionally, the findings of our study may not generalise especially well to males, as the sample was majority (75.5%) female.

There was a possibility of a detection bias. Simply being part of a PA study may have induced participants to feel more motivated for PA, and it is possible that participants could have identified their control group status. This could account for the change in PA level in the control group. Future studies may consider only mentioning health messaging in recruitment.

Finally, the study relied on subjective self-report measures of PA. Motivation outcomes were assessed using a small number of questions, adapted from BREQ-3 [26], which may limit representation of an individual’s overall motivation for activity.

### 4.6. Implications and Future Research

The results of this study show that both GF and LF messages about PA and mental health are effective in further motivating an already active cohort. Future research may explore differential effects by baseline activity status.

Framed messages regarding mental health outcomes of PA via social media could be used for effective PA promotion in universities. This has the potential to be a cost-effective method of mass dissemination [20]. It may also promote a wider impact on the future health of the university population, at a critical age for the establishment of PA behaviours [6]. While this study presents a relationship between messaging and motivation for PA, further research needs to be undertaken to see if increased motivation translates to increased PA levels to improve health, over a longer time period.

It would be beneficial for future studies to include an objective measure to validate the results of self-report data. Objective measurements and long-term follow-up are rarely included in message framing studies and should be attempted [30]. Future research is also required to assess moderators of behaviour change, such as psychological theories for example SDT [9].

## 5. Conclusions

GF and LF messages about PA and mental health outcomes delivered through Facebook were equally effective in increasing motivation for PA, exercise, and active travel in an active university student cohort. However, increased motivation did not translate into detectable differences in PA level in GF and LF groups compared to the control group in this five-week intervention. Framed messages regarding mental health outcomes of PA via social media could be effective for PA motivation in universities.

## Figures and Tables

**Figure 1 ijerph-18-08671-f001:**
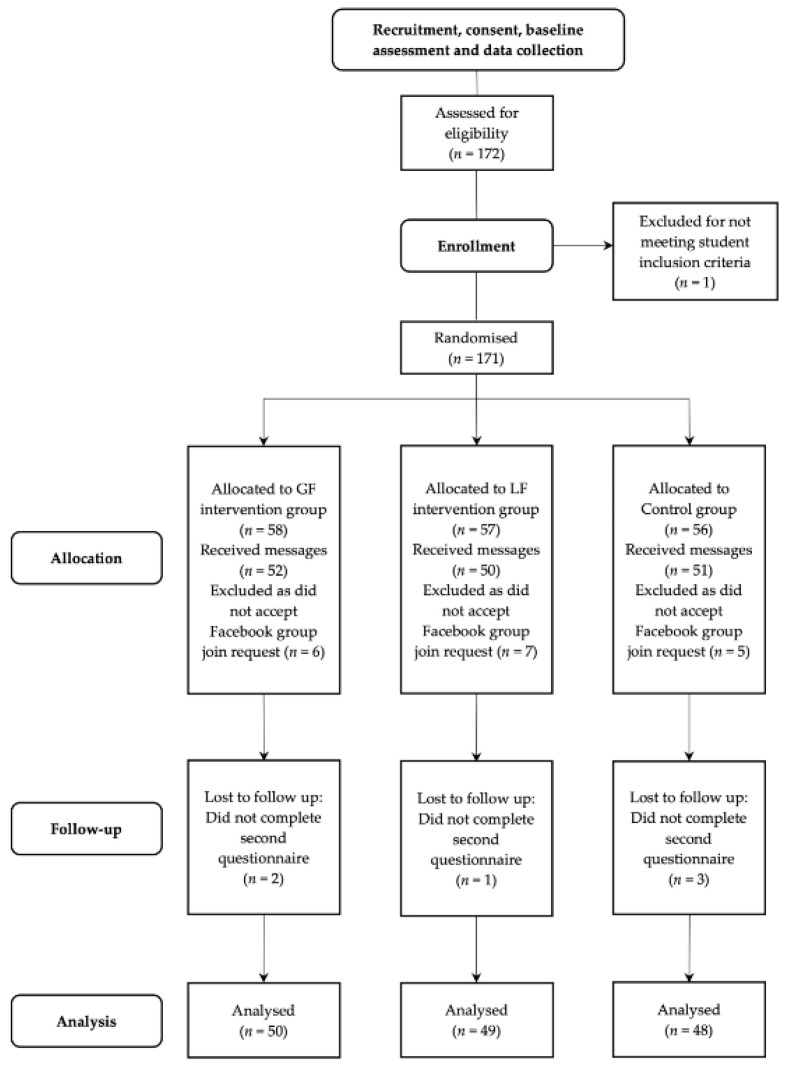
Consort study flow chart displaying the recruitment, intervention, and analysis process of the Facebook messaging intervention.

**Figure 2 ijerph-18-08671-f002:**
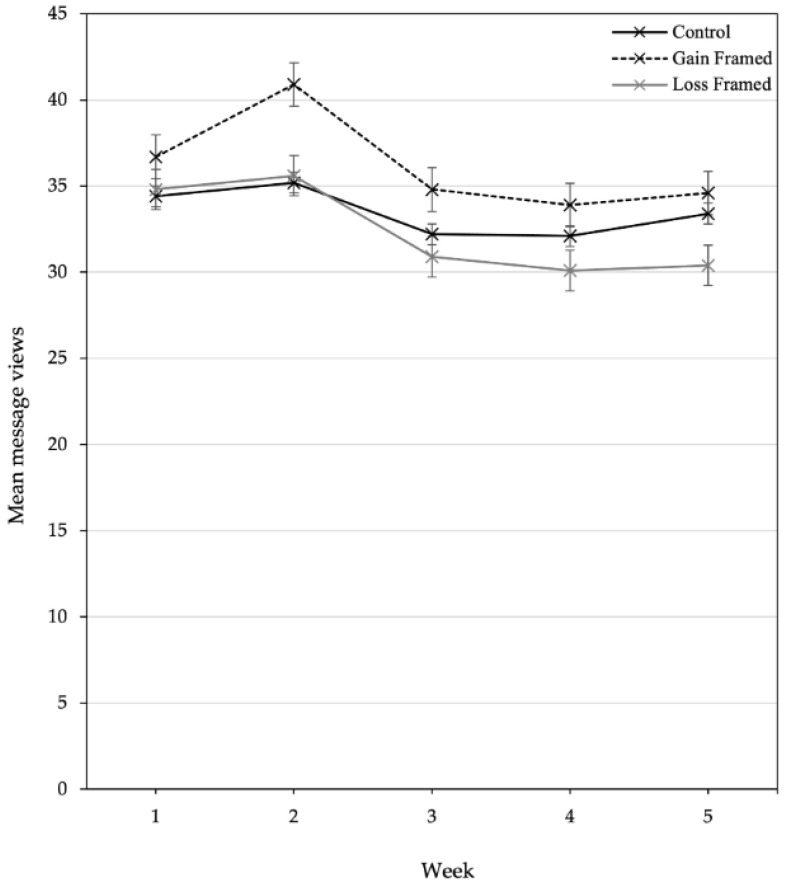
Line graph comparing engagement with messages in the Control, GF and LF Facebook groups, displayed as mean message views per week with standard error bars.

**Table 1 ijerph-18-08671-t001:** Examples of gain-framed, loss-framed and control messages delivered throughout the intervention on separate Facebook groups.

Gain-Framed	Loss-Framed	Control
Aerobic exercises (e.g., running, cycling) are associated with a reduction in social anxiety.	Not taking part in aerobic exercises (e.g., running, cycling) means that you may miss out on the associated reduction in social anxiety.	Your heart beats about 100,000 times each day!
Improvements in self-concept and global self-esteem have been recognised when you participate in regular physical activity!	Improvements in self-concept and global self-esteem may be missed if you do not participate in regular physical activity!	It takes 66 days to form a habit.
Strong evidence demonstrates that physical activity reduces the risk of experiencing depression.	Strong evidence demonstrates that not taking part in physical activity means that the risk of experiencing depression is not reduced.	The hardest bone in the human body is the jawbone.

Note: evidence-based content of the gain- and loss-framed messages were adapted from the Physical Activity Guidelines Advisory Committee (2018) report. The style of messages was consistent in all intervention groups.

**Table 2 ijerph-18-08671-t002:** Demographic characteristics of the sample (N = 147), age presented as the mean and standard deviation in parentheses, between-group difference assessed by ANCOVA. Other characteristics are a percentage of the sample, between-group differences assessed by chi-squared cross tabulation.

Characteristic	All (*n* = 147)	Control (*n* = 50)	Gain-Framed (*n* = 50)	Loss-Framed (*n* = 49)	*p*
Age	20.7 (1.5)	20.6 (1.5)	20.7 (1.6)	20.8 (1.3)	0.66
Sex					
Female	75.5	66.7	76.0	81.6	0.23
Male	24.5	33.3	24.0	18.4	
Ethnicity					
Caucasian	88.4	91.7	90.0	83.7	0.43
Other	11.6	8.3	10.0	16.3	
University					
Scotland	85.5	83.3	82.0	91.5	0.36
Rest of UK	14.4	16.7	18.0	8.5	

Note: *p* values represent statistical significance between control, gain and loss-framed groups.

**Table 3 ijerph-18-08671-t003:** Pre-post change in motivation and PA levels: mean, standard deviation, and effect size (β) estimation via ANCOVA.

Outcome	Control (SD)		Gain-Framed (SD)		ANCOVA		Loss-Framed (SD)		ANCOVA	
	Pre	Post	Pre	Post	β	*p*	Pre	Post	β	*p*
Motivation for PA ^	6.1 (1.1)	6.0 (1.1)	6.2 (1.0)	6.5 (0.8)	1.09	0.007	6.3 (0.9)	6.6 (0.5)	1.10	0.005
Motivation for exercise ^	5.4 (1.6)	5.6 (1.5)	5.7 (1.4)	6.3 (0.8)	1.16	0.001	6.0 (1.1)	6.5 (0.6)	1.15	<0.001
Motivation for active travel ^	5.6 (1.5)	5.5 (1.5)	5.5 (1.5)	6.2 (1.0)	1.18	<0.001	5.7 (1.6)	6.3 (1.0)	1.19	<0.001
PA level										
Total MET minutes/week	3142.3 (2199.2)	3601.8 (2866.4)	3245.6 (2115.6)	3600.6 (2356.5)	0.02	0.6	3517.6 (1985.9)	3799.3 (2297.6)	0.01	0.75

^ Beta coefficients are exponentiated due to log transformation.

## Data Availability

Data are not publicly available due to restrictions of the ethical approval.

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
