# Peer review of "Do Framed Mental Health Messages on Social Media Influence University Students’ Motivation for Physical Activity?"

_ijerph, 2021, doi:10.3390/ijerph18168671_

Round 1

Reviewer 1 Report

I want to thank you for the opportunity to review the article entitled “Do framed mental health messages on social media influence university students’ motivation for physical activity?” for International Journal of Environmental Research and Public Health.

The present study aimed to This study aims to investigate the effects on motivation and PA behavior of framed messaging on social media in university students. The manuscript is fairly well written. I see good potential in this manuscript, although there are some changes that would need attention and revising before being considered for publication.

Abstract

  1. Line 24: the conclusion should have added PA motivation, which consisted most of your outcomes. to the results.
  2. You might only could draw a conclusion that framed messages regarding mental health outcomes of PA via social media could be used for effective PA motivation in universities based on the results of your study.

Introduction:

  1. Line 37: What kind of socio-cultural norms in the university lifestyle makes university students physical inactive. Please describe the details about it.
  2. “Rothman and Salovey [11] theorized that…” could be combined with the third paragraph, which continuing introduce the research about GF and LF message.
  3. Line 49: “Multiple systematic reviews also support this theory in the context of promoting PA [8, 9], however the conclusions are limited by the low methodological quality and heterogeneity of the reviewed literature”.
  4. Low methodological quality and heterogeneity might be the drawbacks of many literature reviews. Is there any original researches which were similar to your study and what’s the limitations of those researches?
  5. Why you target in mental health massages to promote PA, I didn’t find answers in the introduction parts.
  6. What’s the purpose of the sixth paragraph (line 59)? It seems that you haven’t complete this paragraph. Facebook has been promoted for use in health interventions due to previous success in attracting and retaining participants, what’s the relationship with your study, you need to further explain.
  7. I suggest the authors rewrite the introduction part and describe the introduction part from the perspective of different gaps in previous studies.

Methods:

  1. Why you conduct a “5-week” intervention? Is there any reference to cite?
  2. What methods were used to confirm whether they see message on Facebook?
  3. The authors should describe the measurement of outcomes in details, such as: the reference of the motivation questionnaire; how many items of the questionnaire; the reliability of the questionnaire, etc.

Discussion:

  1. Please add more references and discussion on 4.2 (line 199) according to each results.

Conclusion:

14.   You might only could draw a conclusion that framed messages regarding mental health outcomes of PA via social media could be used for effective PA motivation in universities based on the results of your study.

Reviewer 2 Report

The article contains some limitations in the theoretical framework which supports the development of the discourse.

The authors should dedicate a section of the theoretical framework to further explore the role of motivation for PA. It should be noted that the global and systemic concept of health includes aspects referring to the physiological (the one addressed in this article), emotional (linked to motivation) and relational dimensions.

The model followed in identifying three components of motivation for AP should also be explained.

In the methodology section, the authors are asked to revise the title of section 2.2 Development. Perhaps Instruments could be better.

In addition, more information should be given about the questionnaire that was administered to the participants. It would be interesting to know whether the questionnaire identifies different factors or dimensions according to the theoretical framework used. In this section it is essential to mention that a questionnaire adapted from The Behavioural Regulation in Exercise Questionnaire 3 (BREQ-3) has been used. It should be justified why it was adapted and if all factors of this questionnaire were kept.

In the data analysis, authors are asked to consider incorporating effect sizes. If this is not possible, they should reason their response.

A limitation of the article is to present only the relationship between messages received and motivation around health. However, this does not mean that people have changed the level of activity to improve their holistic health. It would have been of interest to complement this study with another fieldwork in which participants could take part in an intervention programme to improve their health. In that context participants could receive messages to influence their motivation.

Round 2

Reviewer 1 Report

It can be accepted now.

Reviewer 2 Report

This new version has improved its content.
Congratulations to the authors for this work.